# Assessment of neurovascular coupling and cortical spreading depression in mixed mouse models of atherosclerosis and Alzheimer's disease

Osman Shabir[1,2,3]*, Ben Pendry[4], Llywelyn Lee[3,5], Beth Eyre[3,5], Paul S Sharp[6], Monica A Rebollar[3,4], David Drew[1], Clare Howarth[2,3,5], Paul R Heath[4], Stephen B Wharton[3,4], Sheila E Francis[1,2,3]†, Jason Berwick[2,3,5]*†

[1]Department of Infection, Immunity and Cardiovascular Disease (IICD), University of Sheffield Medical School, Royal Hallamshire Hospital, Sheffield, United Kingdom; [2]Healthy Lifespan Institute (HELSI), University of Sheffield, Sheffield, United Kingdom; [3]Neuroscience Institute, University of Sheffield, Sheffield, United Kingdom; [4]Sheffield Institute for Translational Neuroscience (SITraN), University of Sheffield, Sheffield, United Kingdom; [5]Sheffield Neurovascular Lab, Department of Psychology, University of Sheffield, Sheffield, United Kingdom; [6]Medicines Discovery Catapult, Alderley Edge, United Kingdom

**\*For correspondence:**
o.shabir@sheffield.ac.uk (OS);
j.berwick@sheffield.ac.uk (JB)

†These authors contributed equally to this work

**Competing interest:** The authors declare that no competing interests exist.

**Abstract** Neurovascular coupling is a critical brain mechanism whereby changes to blood flow accompany localised neural activity. The breakdown of neurovascular coupling is linked to the development and progression of several neurological conditions including dementia. In this study, we examined cortical haemodynamics in mouse preparations that modelled Alzheimer's disease (J20-AD) and atherosclerosis (PCSK9-ATH) between 9 and 12 m of age. We report novel findings with atherosclerosis where neurovascular decline is characterised by significantly reduced blood volume, altered levels of oxyhaemoglobin and deoxyhaemoglobin, in addition to global neuroinflammation. In the comorbid mixed model (J20-PCSK9-MIX), we report a 3 x increase in hippocampal amyloid-beta plaques. A key finding was that cortical spreading depression (CSD) due to electrode insertion into the brain was worse in the diseased animals and led to a prolonged period of hypoxia. These findings suggest that systemic atherosclerosis can be detrimental to neurovascular health and that having cardiovascular comorbidities can exacerbate pre-existing Alzheimer's-related amyloid-plaques.

## Editor's evaluation

In their manuscript, Shabir et al., examine changes in cerebral neurovascular coupling in mouse models of familial Alzheimer disease, atherosclerosis, and a combined comorbidity model to determine the impact of Alzheimer's disease and arteriosclerosis comorbidity on neurovascular coupling. The authors report a set of observations derived from intrinsic optical imaging and multi unit recordings performed in these mouse lines under different combinations of stimulus length and partial oxygen pressure. The discovery that both sensory-driven and injury-based changes in cerebral blood flow (CBF) are perturbed in the settings of Alzheimer's disease and atherosclerosis will help to understand how these diseases impair neurovascular coupling.

## Introduction

Alzheimer's disease (AD) is the most common form of dementia worldwide, with the vast majority of cases being sporadic and occurring at age 65 years and over. Population-based studies have shown that AD and vascular pathologies commonly coexist in the brains of elderly individuals (*Kapasi et al., 2017*; *Matthews et al., 2009*; *Neuropathology Group. Medical Research Council Cognitive and Aging, 2001*; *Rahimi and Kovacs, 2014*). A major cardiovascular pathology that affects as many as up to 60% of all individuals after the age of 55 is atherosclerosis. Atherosclerosis is the progressive thickening, hardening and narrowing of major arteries, including those that supply the brain, such as the carotids (*Lusis, 2000*). Intracranial atherosclerosis does not occur until much later in life, around 75 years and above. As such, Alzheimer's disease that begins around the 8th decade of life is usually present with other comorbidities such as atherosclerosis (*Napoli et al., 1999*). There is also evidence that, not only do these often exist as comorbidities, but they may interact pathogenically with vascular disease and neurovascular unit changes contributing to AD (*Iadecola, 2017*; *Kapasi and Schneider, 2016*). To date, there are very limited models of comorbidity with respect to preclinical studies, and instead models have been very specific and 'pure', and not reflective of the clinical pathology in humans. Atherosclerosis is known to be a major risk factor for the development of dementia. The progressive atheromatous plaque build-up within cerebral arteries that supply the cortex over time can lead to stenosis producing insufficient oxygen delivery to the brain parenchyma, potentially resulting in neuronal death and symptoms of dementia (*Shabir et al., 2018*). Indeed, the vascular cognitive impairment (VCI) which precedes the onset of dementia may be attributed to a variety of different vascular pathologies affecting either systemic or intracranial vasculature (both large or small vessels) (*Iadecola et al., 2019*). Due to the complexity of atherosclerosis and dementia pathogenesis, understanding the mechanisms of their mutual interactions is necessary if efforts to develop therapeutics to prevent VCI and vascular dementia, which currently has no disease-modifying cure, are to succeed.

The breakdown of NVC is thought to be an important and early pathogenic mechanism in the onset and progression of a range of neurological conditions (*Zlokovic, 2011*). In the present study, we aimed to investigate neurovascular function in mid-aged (9–12 m old) mice where atherosclerosis was a comorbidity. We used a novel model of atherosclerosis that utilises a single adeno-associated virus (AAV) i.v. injection of a gain-of-function mutation (D377Y) to proprotein convertase subtilisin/kexin type 9 (rAAV8-mPCSK9-D377Y), combined with a high-fat Western diet to induce atherosclerosis in most adult mouse strains (*Bjørklund et al., 2014*; *Roche-Molina et al., 2015*). This leads to the constitutively active inhibition of the LDL-receptor preventing cholesterol internalisation and degradation by hepatocytes, leading to hypercholesterolaemia to occur and the development of robust atherosclerotic lesions within 6–8 weeks (*Bjørklund et al., 2014*). Furthermore, in order to address the effect atherosclerosis could have on mild Alzheimer's pathology, we combined the atherosclerosis with the mild J20-hAPP mouse model of familial Alzheimer's disease (fAD) to create a mixed comorbid mouse model (J20-PCSK9-MIX). The J20-hAPP mouse model of fAD over-expresses human amyloid precursor protein (hAPP) with the Swedish (K670N and M671L) and the Indiana (V7171F) familial mutations (*Mucke et al., 2000*), which begin to develop amyloid-beta (Aβ) plaques around 5–6 months of age, and show signs of cognitive impairments from 4 months (*Ameen-Ali et al., 2019*). We hypothesised that atherosclerosis would exacerbate Alzheimer's disease pathology in the brain and that neurovascular function would be further worsened compared to AD or ATH models alone. We have previously reported no significant alterations to evoked-haemodynamics in the J20-AD model of the same age (9–12 m); however, under acute imaging sessions where an electrode was inserted into the brain, we found significantly perturbed haemodynamics (*Sharp et al., 2020*). We hypothesised that electrode insertion causes cortical spreading depression (CSD). Based on recent data linking migraine with aura with cardiovascular disease (*Kurth et al., 2020*), we hypothesised that experimental CSD might be heightened in all disease models. We report that experimentally induced atherosclerosis in the J20-AD model increased the number of Aβ plaques by 300%. Furthermore, experimental CSD is more severe in all diseased groups compared to WT controls.

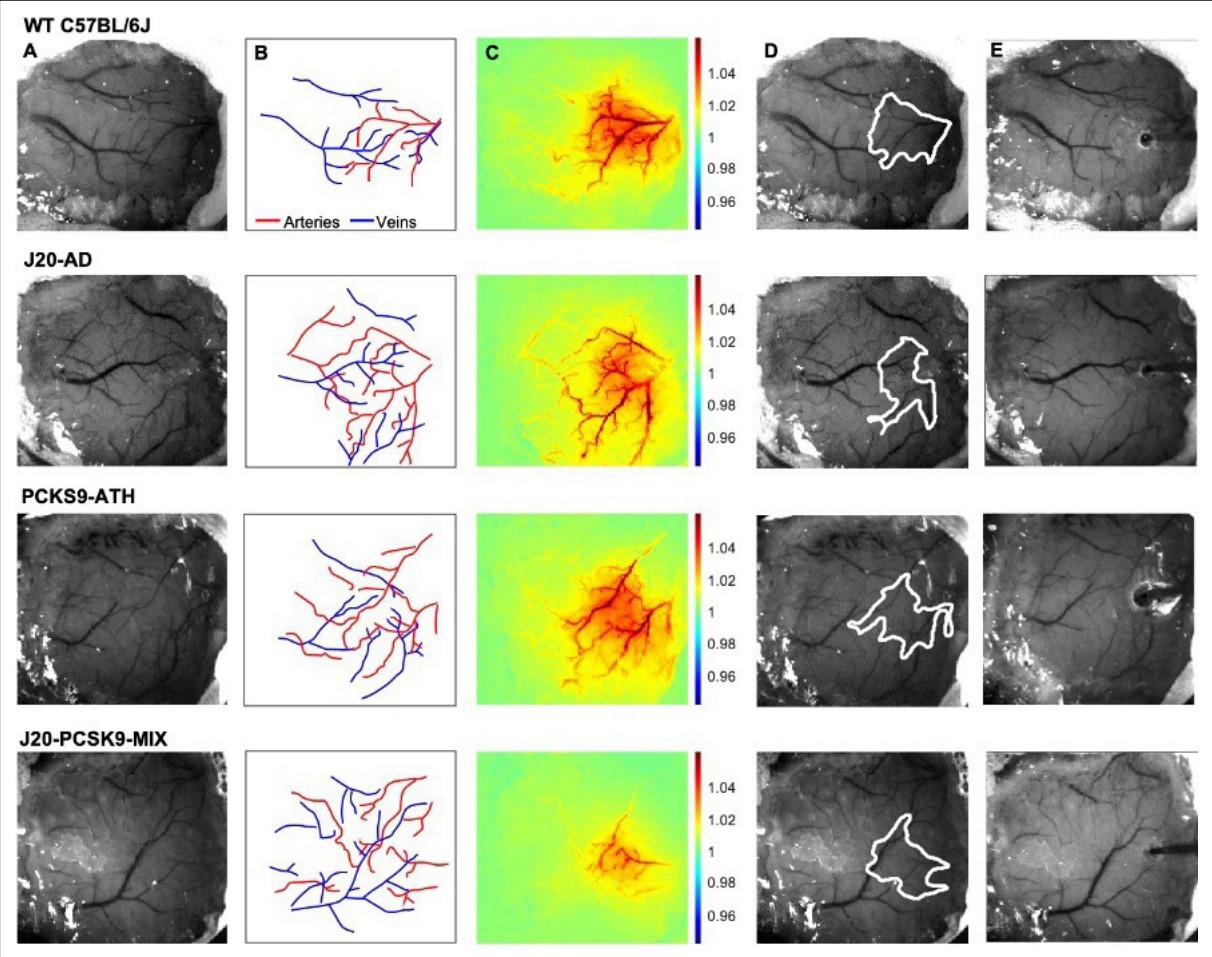

**Figure 1.** Experimental setup and data derivation. (**A**) Raw image of representative thinned cranial windows for WT, J20-AD, PCKS9-ATH AND J20-PCSK9-MIX mice (chronic imaging session). (**B**) Vessel map outlining the major arteries and veins within the thinned cranial window. (**C**) HbT spatial activation map showing fractional changes in HbT in response to a 16s-whisker stimulation. (**D**) Automated computer-generated region of interest (ROI) determined from the HbT activation response in C from which time-series for HbT, HbO, and HbR are generated. (**E**) Raw image of the same animals in terminal acute imaging sessions with multichannel electrodes inserted into the active ROI determined from chronic imaging session.

## Results

### 2D-optical imaging spectroscopy (2D-OIS) measures brain cortical haemodynamics through a thinned cranial window

We performed chronic imaging of the brain cortex 3 weeks post-surgery, where the thinned cranial window remained intact (*Figure 1A/B*), as described previously (*Shabir et al., 2020*; *Sharp et al., 2020*). We deployed a range of stimulations (2 s & 16 s mechanical whisker stimulations) with the mouse breathing both 100% oxygen (hyperoxia) and 21% oxygen (normoxia), in addition to recording transitions between conditions and performing a 10% hypercapnia test to test the maximum dilation of vessels. Each experimental day consisted of the same set of experiments with consistent timings to ensure reliability across all animal groups. First, a 2s-whisker stimulation (5 Hz) with the mouse breathing 100% oxygen; hyperoxia, consisting of 30 trials, second, a 16s-whisker stimulation consisting of 15 trials. Animals were then transitioned from hyperoxia to 21% oxygen; normoxia, and the baseline haemodynamic changes were recorded. The same set of stimulations were deployed under normoxia (2 s & 16 s stimulations), before transitioning back to hyperoxia for a final 10% hypercapnia test. Using these stimulations, activation maps of blood volume; total haemoglobin (HbT), can be generated (*Figure 1C*). Mice were allowed to recover and after 1 week, a final acute imaging session was performed. In this setup, a small burr-hole was drilled through the thinned skull overlying the active region of interest (ROI) as determined from the chronic imaging sessions (*Figure 1D*), and a

multichannel electrode was inserted into the brain (*Figure 1E*) to record neural activity simultaneously. We then imaged and recorded the baseline haemodynamics for a 35-min period to observe the effect electrode insertion, before commencing the first stimulation. This was also done to record baselines on chronic imaging sessions.

## Chronic haemodynamic responses in the brain are reduced in PCSK9-ATH mice

Cortical haemodynamics were imaged through a thinned cranial window to determine whether evoked cortical haemodynamics were different between 9 and 12 m old wild-type (WT), atherosclerotic (PCSK9-ATH), Alzheimer's (J20-AD) & mixed (J20-PCSK9-MIX) mouse models (*Figure 2*). Across all stimulations and conditions, ATH-PCSK9 mice displayed a significant reduction of evoked blood volume responses (HbT; peak value) compared to WT controls. J20-AD mice and J20-PCSK9-MIX mice did not exhibit a significant change in HbT across all stimulation conditions compared to WT mice. Evoked HbT responses; although initially are smaller in J20-PCSK9-MIX mice, recovered to match that of J20-AD mice later in the experimental protocol under normoxia (*Figure 2*). Furthermore, rise times and times to peak for HbT were not significantly different across any of the groups (*Figure 2—figure supplement 1* & *Figure 2—figure supplement 2*). Levels of oxyhaemoglobin (HbO) were significantly reduced in PCSK9-ATH mice but showed a reduced trend in J20-PCSK9-MIX mice too. The washout of deoxyhaemoglobin (HbR) was significantly reduced in PCSK9-ATH mice compared to WT, but also showed a reduced trend across all diseased groups across all conditions compared to WT mice. Examining the shape of the HbR washout in the some of the disease models shows some key differences compared to WT mice. Most notably, J20-PCSK9-MIX mice show a triphasic response with an initial increase in HbR followed by a decrease then return to baseline (*Figure 2*). This contrasts with the biphasic waveform seen in WT, J20-AD and PCSK9-ATH mice; albeit lower in disease mice compared to WT controls. All mice displayed stable and robust haemodynamic responses across the experimental protocol (*Figure 2—figure supplement 3*). Finally, vascular reactivity as determined by the response to 10% hypercapnia was not significantly different between any of the diseased groups (*Figure 2—figure supplement 4*).

## CSD is worse in diseased animals and impacts haemodynamic recovery to baseline

One week after recovery from the chronic imaging protocol, an acute imaging experiment was performed wherein a small-burr hole was drilled into the skull overlying the active region (determined from HbT responses from chronic experiments) and a microelectrode was inserted into the brain to a depth of 1500–1600 μm to obtain neural electrophysiology data in combination with the imaging of cortical haemodynamics by 2D-OIS. Electrode insertion into the brain resulted in a wave of haemodynamic changes that occurred in all mice (CSD) (*Figure 3*). In WT mice, electrode insertion led to a small decrease in HbT (indicative of vasoconstriction) followed by a robust HbT bounce back (indicative of vasodilation), immediately followed by a small, sustained reduction in HbT that persisted for some time (*Figure 3A*, top). In J20-AD mice, electrode insertion caused a large reduction in HbT to occur which spread across the cortex in a strong wave of reduced HbT (indicative of stronger vasoconstriction than in WT mice) that was followed by a very small, attempted recovery. This was masked by a large sustained and prolonged reduction in HbT within contiguous vessels that persisted for some time (*Figure 3A*, bottom). The largest reduction in HbT post-CSD occurred in J20-AD mice, followed by PCSK9-ATH mice, then J20-PCSK9-MIX mice compared to WT controls. The smallest of all CSD occurred in WT mice (*Figure 3B*). A prolonged and sustained reduction in HbT below baseline persisted in all mice post-CSD, however, this effect was recovered to baseline in WT mice during the first stimulation experiment 35 min after the CSD occurred (*Figure 3B*). In all disease mouse models, the constriction below baseline was more severe and persisted for a much longer time with a slower haemodynamic recovery. Following CSD, stimulation-evoked haemodynamic changes were not significantly different in any of the diseased groups overall, although they were initially smaller in the first two stimulations for PCSK9-ATH mice (*Figure 3—figure supplement 1*). This is due to the effect that CSD has on baseline and evoked haemodynamics that then persists for some time up to and during the experimental stimulations. Thus, the acute data presents a different outlook compared to chronic-only imaging sessions that do not have the added confound of CSD.

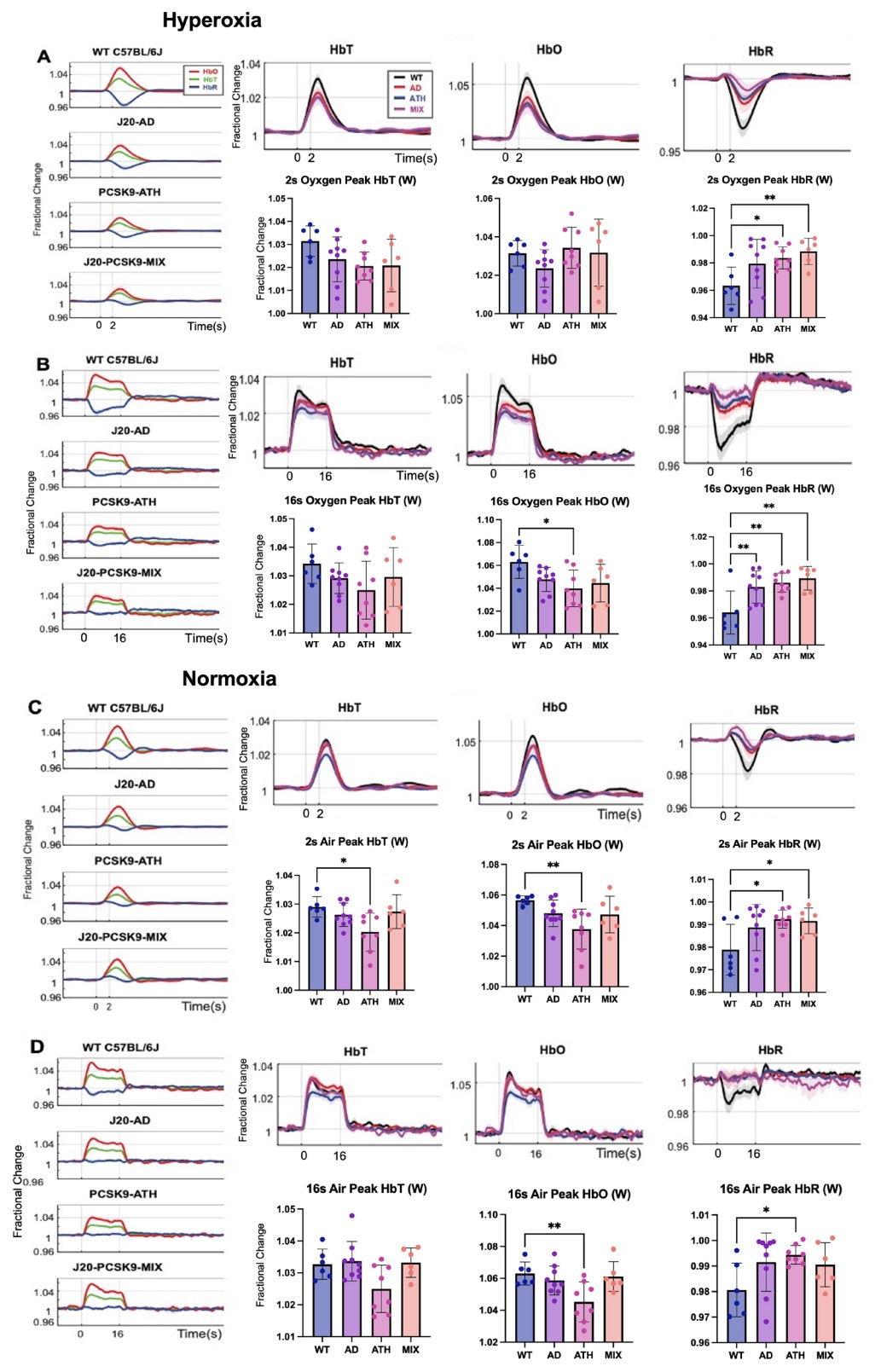

**Figure 2.** Fractional changes in chronic stimulus-evoked haemodynamic responses (Peak/Mean; Whisker Region). (**A**) 2s-stimulation in which all animals were breathing in 100% oxygen (hyperoxia). (**B**) 16-stimulation (hyperoxia). (**C**) 2s-stimulation in which all animals were breathing in 21% oxygen (normoxia). (**D**) 16s-stimulation (normoxia). All animals aged 9–12 m: WT (n = 6), J20-AD (n = 9), PCSK9-ATH (n = 8), J20-PCSK9-MIX (n = 6). (**HbT:**) There was no

*Figure 2 continued on next page*

*Figure 2 continued*

significant overall effect of disease F(3,25)=2.83, p = 0.059. However, Dunnett's (two-sided) multiple comparisons test revealed there was a significant difference between WT and ATH (p = 0.023). As expected, there was a significant effect of experiment, F(1.65,41.14) = 13.64, p < 0.001. There was also no significant interaction effect between experiment and disease, F(4.94,41.14) = 1.50, p = 0.211. (**HbO**): There was a significant overall effect of disease F(3,25)=4.84, p = 0.009. Dunnett's (two-sided) multiple comparisons test revealed there was a significant difference between WT and ATH (p = 0.002). There was a significant effect of experiment, F(1.47,36.72) = 15.348, p < 0.001. There was no significant interaction effect between experiment and disease, F(4.41,36.72) = 1.64, p = 0.181. (**HbR**): There was a significant overall effect of disease F(3,25)=4.86, p = 0.008. Games-Howell multiple comparisons reveal HbR peak is significantly different for WT vs ATH (p = 0.040). There was a significant effect of experiment, F(1.69,42.28) = 17.33, p < 0.001. There was a significant interaction between experiment and disease interaction: F(5.07, 42.28) = 3.19, p = 0.015. All error bars (lightly shaded) are± SEM. Vertical dotted lines indicate start and end of stimulations. Bar graphs compare whisker region evoked HbT, HbO, and HbR within each group to WT mice for each experimental condition. Dots indicate individual animals. One-way ANOVA was performed with post-hoc Dunnett's test to determine significance within each condition denoted as p > 0.05 as * and p > 0.01 as ** (Error bars ± SD).

The online version of this article includes the following figure supplement(s) for figure 2:

**Figure supplement 1.** Time to peak (chronic).

**Figure supplement 2.** Rise times (chronic).

**Figure supplement 3.** Concatenated data showing stability and robustness of the mouse imaging preparation.

**Figure supplement 4.** Chronic (left) and acute (right) hypercapnia.

## Stimulus-evoked neural activity is not significantly altered in any disease groups compared to WT mice

In the final imaging session and after a 35-min period of recovery post-electrode insertion, the first experimental stimulation was performed (2s-stimulation in 100% oxygen) where evoked cortical haemodynamics were imaged simultaneously with the recording of neural multi-unit activity (MUA). Evoked-MUA response were not significantly different in any of the diseased groups compared to WT mice (*Figure 4*), suggesting that the significantly different evoked-HbT in PCSK9-ATH mice (observed on chronic imaging sessions) was due to neurovascular breakdown. Initially, the MUA was slightly lower for J20-AD, PCSK9-ATH and J20-PCSK9-MIX mice compared to WT mice (*Figure 4A*), however, later in the experimental session by the last stimulation, there was no observable difference in MUA between any of the groups (*Figure 4*). Thus, this suggests that the neural MUA maybe initially smaller after the CSD had occurred, however, recovered fully with time. The haemodynamic responses in the acute experimental session were not significantly different across all stimulations for

**Table 1.** Summary of statistics.

| Metric | Statistical test | p-Values and summary |
|---|---|---|
| Neural multi-unit activity | Two-way ANOVA post hoc Dunnett's | *Not significant for any model.*<br>Peak: F(3,24)=2.24, p = 0.109 not significant<br>AUC: F(3,24)=1.66, p = 0.114 not significant |
| Peak arterial region chronic haemodynamics | Two-way ANOVA post hoc Dunnett's | **HbT WT vs ATH p = 0.026(*).**<br>*No other metric is significant for peak arterial data across all stimulations/conditions.* |
| Chronic hypercapnia | One-way ANOVA post hoc Dunnett's<br>Kruskal Wallis (AUC) | *Not significant for any model.*<br>Mean HbT F(3,12.06) = 0.49, p = 0.694<br>Arterial mean HbT *F* = 0.692, p = 0.566<br>AUC HbT H(3)=4.011, p = 0.26 |
| Acute hypercapnia | One-way ANOVA post hoc Dunnett's<br>Kruskal Wallis (Mean Arterial HbT) | *Not significant for any model.*<br>Mean HbT F(3,24)=0.775, p = 0.519<br>AUC HbT F(3,24)=0.78, p = 0.519<br>Arterial mean HbT H(3)=3.6, p = 0.308 |
| Amyloid-beta staining | Unpaired T-tests | Hippocampal: p = 0.036(*)<br>Cortical: p = 0.337(ns)<br>Whole Brain: p = 0.0328(*) |

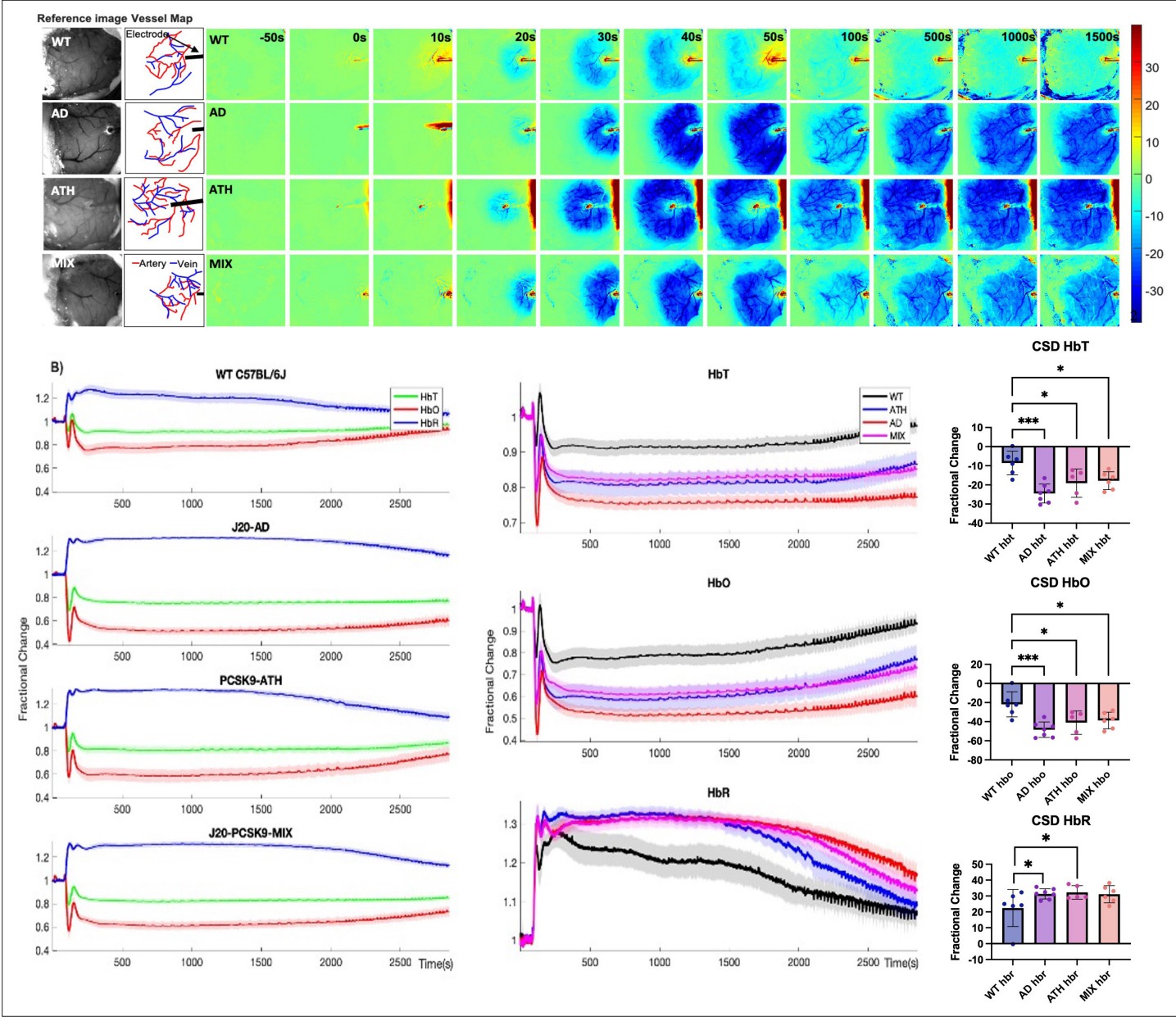

**Figure 3.** Cortical spreading depression (CSD) in WT, diseased and comorbid animals. (**A**) Representative montage time-series of WT, J20-AD, PCSK9-ATH and J20-PCSK9-MIX mice showing HbT changes post-electrode insertion. Electrode insertion occurs at t = 0 s. Colour bar represents percentage changes in HbT from baseline. (**B**) *Left:* Average CSD haemodynamics profiles for control animals (WT C57BL/6 J and nNOS-ChR2) (n = 7); *mean HbT (625–1250 s) C57BL/6 J (n = 3): 0.9031 ± 0.028598 STD, mean HbT nNOS-ChR2 (n = 4): 0.92355 ± 0.063491 STD (p = 0.64816; ns),* J20-AD (n = 7), PCSK9-ATH (n = 5) and J20-PCSK9-MIX (n = 6) mice. *Right:* Averaged changes to HbT (top), HbO (middle) and HbR (bottom) upon CSD in the different mouse groups. (HbT:) A one-way ANOVA showed significant effect of disease for HbT (F(3,21)=9.62, p < 0.001). Dunnett's two-sided multiple comparisons showed that AD vs WT p < 0.001, ATH vs WT p = 0.012 & MIX vs WT p = 0.020. (HbO:) one-way ANOVA showed significant effect of disease for HbO (F(3,21)=8.51, p = 0.001). Dunnett's two-sided multiple comparisons showed that AD vs WT p < 0.001, ATH vs WT p = 0.01 and MIX vs WT p = 0.017. (HbR:) one-way ANOVA showed significant effect of disease for HbR (F(3,21)=3.60, p = 0.031). Dunnett's two-sided multiple comparisons showed that AD vs WT p = 0.037, ATH vs WT p = 0.038, MIX vs WT p = 0.053. All error bars (lightly shaded) are± SEM. Bar graphs illustrate HbT, HbO, and HbR fractional changes between 625 and 1250 s with one-way ANOVA showing significant effect of disease for all parameters. Post-hoc Dunnett's multiple comparisons are denoted by asterixis. Error bars ± SD.

The online version of this article includes the following figure supplement(s) for figure 3:

**Figure supplement 1.** Fractional changes in acute stimulus-evoked haemodynamic responses.

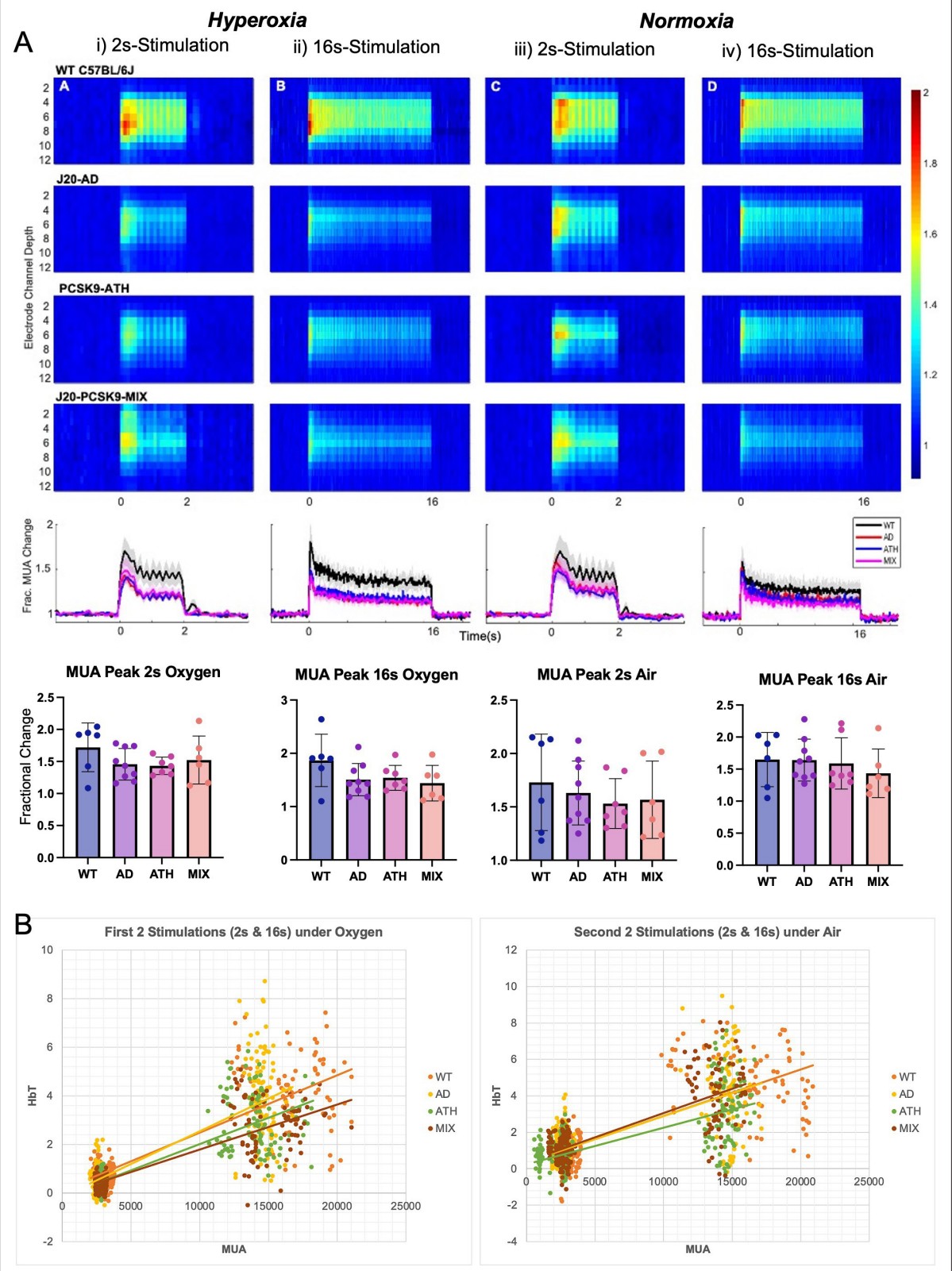

**Figure 4.** Evoked neural multi-unit activity (MUA) responses. (**A**) MUA heat maps showing fractional changes in MUA along the depth of the cortex (channels 4–8) in response to stimulations in WT C57BL/6 J (n = 6), J20-AD (n = 9), PCSK9-ATH (n = 7) and J20-PCSK9-MIX (n = 6) mice. Overall effect of disease on MUA F(3,24)=2.24, p = 0.109 (2-way mixed design ANOVA). There was a significant effect of experiment, as expected, F(2.26, 54.16) = 6.83, p = 0.002. There was no significant interaction between experiment and disease F(6.77, 54.16) = 0.70, p = 0.670. All error bars (lightly shaded) are±

*Figure 4 continued on next page*

*Figure 4 continued*

SEM. Bar graphs showing fractional changes in peak MUA for each group in each experimental condition with one-way ANOVA showing no significant differences in MUA between groups for any experiment. Error bars ± SD. (**B**) Neurovascular correlation plots of trial-by-trial data (not individual animal means) comparing evoked MUA (peak) against subsequent evoked HbT (peak) changes. The two plots are time related with the left being the first two set of stimulations after electrode insertion/CSD in which animals were breathing in 100% oxygen (hyperoxia) and the right being the second set of stimulations occurring later in the experimental paradigm in which all animals were breathing in air (21% oxygen; normoxia). The 2s-stimulations are the smaller cluster to the left of each plot whereas the 16s-stimulations are the larger cluster to the right of each plot. Line equations and R-values (regression analysis). *Oxygen (Hyperoxia).* WT: y = 0.0002x + 0.1194, R² = 0.6867. AD: y = 0.0003 x-0.1467, R² = 0.6693. ATH: y = 0.0002 x-0.1446, R² = 0.6936. MIX: y = 0.0002 x-0.0659, R² = 0.7044. *Air (Normoxia).* WT: y = 0.0003x + 0.3501, R² = 0.5736. AD: y = 0.0003x + 0.3502, R² = 0.5066. ATH: y = 0.0002x + 0.251, R² = 0.4898. MIX: y = 0.0003x + 0.3739, R² = 0.5996.

any of the diseased groups. Neurovascular coupling was assessed by producing a correlation plot comparing evoked MUA with the associated evoked HbT changes across all stimulations and groups to determine if there was a genuine neurovascular deficit (*Figure 4B*). Neurovascular coupling is impaired in PCSK9-ATH mice as per given evoked MUA there is less evoked HbT in PCSK9-ATH mice compared to WT controls (*Figure 4B*) despite no significant differences in vascular reactivity (as assessed by hypercapnia; *Figure 2—figure supplement 4*). This suggests a functional or metabolic deficit.

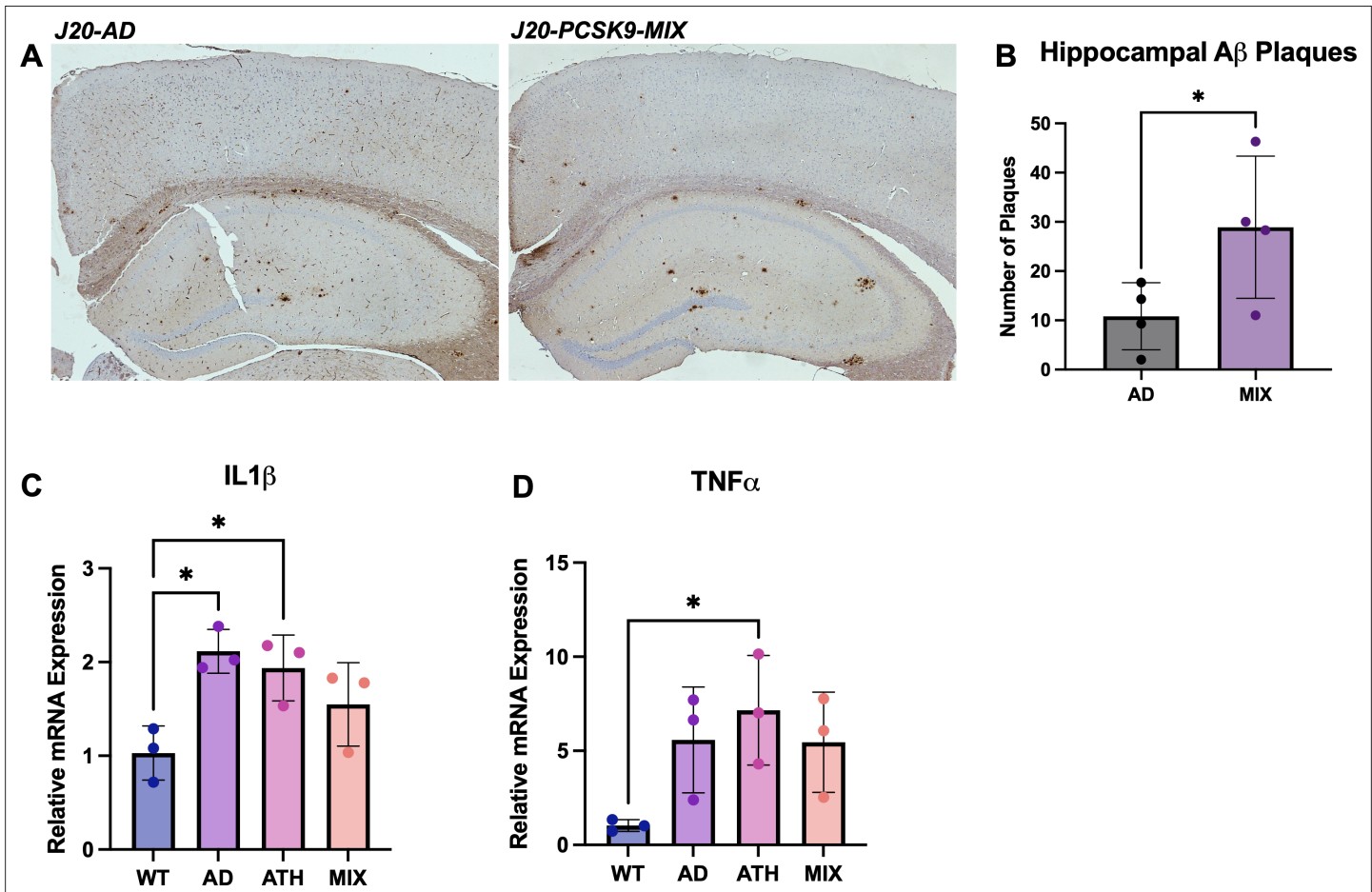

**Figure 5.** Neuropathology and Neuroinflammation. (**A**) Representative histological coronal hippocampal sections for J20-AD and J20-PCSK9-MIX mice stained with anti-Aβ to visualise Aβ plaques. (**B**) Increased number of amyloid-beta plaques in the hippocampus of J20-PCSK9-MIX mice compared to J20-AD mice (p = 0.036; unpaired t-test) (n = 4 each). Cortical plaques p = 0.3372 (data not shown). (**D**) qRT-PCR for 1L1β: AD vs WT p = 0.011, ATH vs WT p = 0.0278, MIX vs WT p = 0.218 (one-way ANOVA with post-hoc Dunnett's multiple comparisons test). (**E**) qRT-PCR for TNFα: AD vs WT p = 0.1197, ATH vs WT p = 0.0370, MIX vs WT p = 0.1313 with post-hoc Dunnett's multiple comparisons test. All error bars are± SD.

## Increased number of hippocampal Aβ plaques in J20-PCSK9-MIX mice. Increased neuroinflammation in J20-AD and PCSK9-ATH mice

Immunohistochemistry was performed on J20-AD and J20-PCSJK9-MIX mice to assess whether there were any specific differences in AD neuropathology changes. Staining was performed for Aβ plaques, and these were quantified within the hippocampus and the cortex. Aβ plaques were significantly increased by threefold in the hippocampi of J20-PCSK9-MIX mice compared to J20-AD mice (*Figure 5A/B*). Next, neuroinflammation was assessed by qRT-PCR for two key inflammatory markers: interleukin-1β (IL1β) and tumour necrosis factor-α (TNFα) to assess the degree of neuroinflammation present globally within the brain. IL1β mRNA was significantly upregulated in J20-AD and PCSK9-ATH mice (*Figure 5C*). TNFα mRNA was significantly upregulated in PCSK9-ATH mice only (*Figure 5D*). J20-PCSK9-MIX mice displayed the lowest inflammatory changes in IL1β & TNFα compared to the other diseased groups, though this was still higher than WT mice.

## Discussion

The present study investigated neurovascular function in a novel experimental model of atherosclerosis (PCSK9-ATH) and for the first time, in a comorbid setting whereby atherosclerosis was experimentally induced in a well characterised model of AD; J20-hAPP, to create a mixed comorbid model (J20-PCSK9-MIX). These mice were compared to age-matched (9–12 m) WT C57BL/6 J controls, and J20-AD mice. Given that systemic atherosclerosis is a major risk factor for dementia, the mechanisms underpinning the relationship between atherosclerosis, neurovascular decline and dementia are still largely unclear.

In the first part of the study, we characterised evoked-haemodynamic responses using a chronic skull-intact and surgery-recovered mouse preparation. We found that PCSK9-ATH mice displayed significantly reduced evoked blood volume (HbT) responses, in addition to reduced levels of oxyhaemoglobin (HbO) and notably, an impaired washout of deoxyhaemoglobin (HbR) across all stimulations and conditions. The J20-PCSK9-MIX mice did not display a significant reduction in HbT, nor in HbO or HbR levels but did trend towards a reduction in both. Interestingly, J20-PCSK9-MIX mice displayed a triphasic HbR profile compared to the typical biphasic shape seen in the other groups, with an initial inverted response. This could reflect worse overall metabolic deficits despite functional neurovascular coupling. With respect to J20-AD mice, we did not observe any significant alterations to HbT as previously published (*Sharp et al., 2020*). Another important finding from the present study was that 10%-hypercapnia responses were not significantly different in any of the mice compared to WT controls (*Figure 2—figure supplement 4*), thus suggesting that vascular reactivity was not impaired in any of the mice, indicating that cerebral arterioles were unaffected by atherosclerosis at this timepoint (9–12 m). Thus, the basis of reduced HbT in PCSK9-ATH mice cannot be attributed to intracranial atherosclerosis. A recent study corroborated our findings showing that cerebrovascular reactivity and cerebral blood flow was preserved in the Tg-SwDI model of Alzheimer's (*Munting et al., 2021*), similar to our J20-model.

In the second part of the study, we obtained neural multi-unit activity (MUA) by inserting a multi-channel electrode into the active region defined from the chronic imaging experiments. As we showed in our previous reports (*Shabir et al., 2020*; *Sharp et al., 2020*), the technical procedure of electrode insertion causes a cortical spreading depression (CSD) to occur in all animals. Here, we describe the CSD and its recovery on the different disease groups. CSD has two distinct phases: (1) a wave of depolarisation within the grey matter characterised by neuronal distortion leading to a large change of the membrane potential whereby neuronal activity is silenced (spreading depression) and (2) haemodynamic changes that accompany neuronal spreading depolarisation which typically result in a wave of prolonged reduced perfusion that persists for some time (*Ayata and Lauritzen, 2015*; *Dreier, 2011*). CSD does not typically occur in healthy brain tissue, however, it is a common neurophysiological occurrence in certain pathological conditions including migraine, epilepsy, brain injury, hyperthermia, hypoxia, and ischaemia (*Dreier, 2011*). In WT mice, the initial reduction in HbT is small, and a robust haemodynamic recovery occurs which allows for neurovascular coupling to occur to sustain neurons metabolically. This is a marked difference to the diseased animals, which upon electrode insertion to cause a CSD, exhibit a large reduction in HbT with an extremely limited haemodynamic recovery resulting in a prolonged constriction, severe reductions to blood volume and

HbO and HbR levels indicating hypoxia and ischaemia. We would also expect to see increased basal levels of ischaemia and hypoxia to some extent in the disease models, particularly the mixed model. Investigation of such markers will form the basis of future studies. Thus, the haemodynamic response to CSD in diseased mice is severely inappropriate and can lead to long lasting devastating effects such as widespread cortical pannecrosis of neurons and astrocytes (*Dreier, 2011*). As our data show, baseline blood volumes do not recover in the diseased animals for a much longer period compared to WT animals, with the most profound CSD occurring in J20-AD mice, followed by PCSK9-ATH mice.

CSD may be the neuropathological link between migraine, stroke, cardiovascular disease and dementia in which cardiovascular risk factors, genetics and other lifestyle factors which prime the onset of migraine to occur lead to vascular vulnerability within the brain predisposing affected individuals to an increased risk of cerebral ischaemia and haemorrhagic stroke (*Ripa et al., 2015*). There is accumulating evidence to suggest that shared genetic and associated clinical features observed in migraine patients are involved in the increased vulnerability to cerebral ischaemia, therefore, predisposing affected individuals to stroke and white matter lesions associated with dementia (*Yemisci and Eikermann-Haerter, 2019*). The underlying mechanism being CSD; the neurophysiological feature of aura in migraines, whose induction threshold can be reduced by genetic mutations and systemic comorbidities that contribute to vascular dysfunction and neuroinflammation (*Yemisci and Eikermann-Haerter, 2019*). Indeed, mouse models of cerebral autosomal dominant arteriopathy with subcortical infarcts and leukoencephalopathy syndrome (CADASIL); a genetic cerebrovascular disease caused by *NOTCH3* mutations that has a high frequency of migraines with aura, have enhanced CSD linking a dysfunctional neurovascular unit with migraine with aura (*Eikermann-Haerter et al., 2011*). Furthermore, a recent study examined women who suffered from migraines with and without aura and found that those that suffered migraines with aura had a higher incidence rate of cardiovascular disease compared to women without aura or any migraines (*Kurth et al., 2020*). In addition, another recent study found that migraine history was positively associated with an increased risk of developing both all-cause dementia and AD, but not VaD (*Morton et al., 2019*). Our study, along with the previously discussed studies provide an explanation for these recent findings and highlights how systemic disease can prime the brain to allow profound CSDs to occur in the context of migraine, and as such, migraine frequency and intensity may be related to the onset of neurological disease by later in life including dementia.

Neural MUA data; generated by local neuronal action potentials, was not significantly altered across any of the stimulations or conditions for any of the disease groups compared to WT controls (*Figure 4A*), although it does appear that there may be a trend toward slightly higher MUA for WT mice. A consistent finding irrespective of stimulation and condition was that PCSK9-ATH mice display consistently reduced evoked-HbT responses against evoked MUA (observed in chronic and acute experiments respectively) compared to WT controls, which suggests an advanced level of neurovascular breakdown and inefficiency (*Figure 4B*). This is further supported by the whisker stimulation data while animals breathed air (21% oxygen) where HbR is inverted in PCSK9-ATH and J20-PCSK9-ATH mice (*Figure 2*); as such, when breathing air the mice could be hypoxic and have a metabolic deficit. Other groups have found similarly reduced blood flow in the ApoE$^{-/-}$ model without altered cortical activation (*Ayata et al., 2013*). A recent study found decreased tissue oxygenation in the LDLR$^{-/-}$ mouse model of atherosclerosis (*Li et al., 2019*), and this is most likely to be the case in the PCSK9 model. Initially, the effect of CSD confounded the neurovascular assessment for both J20-AD and J20-PCSK9-MIX mice. Haemodynamic responses to the first 2 s and 16 s stimulations were altered in these mice, as compared to WT. However, haemodynamic responses to later stimulations matched those of WT mice, confirming the non-significant effect of disease. Throughout the whole experimental period, neurovascular measures in PCSK9-ATH mice remained consistently impaired compared to WT (*Figure 4B*). Together, these data indicate that whilst CSD has a profound effect on stimulation-evoked haemodynamic responses in all disease groups, time allows for recovery for WT, J20-AD and J20-PCSK9-MIX groups, but PCSK9-ATH; despite being recovered, are still affected due to genuine long-lasting neurovascular deficits.

A question that arises is why the J20-PCSK9-MIX mice HbT responses are not more severely impaired than J20-AD and PCSK9-ATH? There may be redundancies that occur physiologically to compensate for mild hypoxia in the brain, such as the possible angiogenesis within the brain. Angiogenesis is known to be triggered in cerebral microvessels in AD in response to increased Aβ and

neuroinflammation and may initially reflect a compensatory mechanism to increase perfusion (*Jefferies et al., 2013*). In addition, the levels of neuroinflammation seen in these mice may be due to an altered disease-course and examining temporospatial expression may reveal much higher levels of inflammation in this mixed model at an earlier time-point. Other markers of inflammation may be upregulated compared to those that we assessed, and future studies would incorporate transcriptomic approaches to identify other mechanisms or markers. Nevertheless, a key translational finding from our study was that J20-PCSK9-MIX mice displayed a significant increase in the number of hippocampal plaques.

There are several notable limitations with the present study. Firstly, all imaging was performed on lightly anaesthetised animals, which is known to compromise neurovascular function (*Gao et al., 2017*). However, previous research from our laboratory has developed an anaesthetic regimen that is comparable to awake imaging in terms of the haemodynamic responses to physiological whisker stimulation with little effect on vascular reactivity (*Sharp et al., 2015*). The benefits of lightly anaesthetised preparations over awake preparations is that we can avoid the multiple considerations of behavioural state in which the animals may be whisking, grooming as well as their arousal and stress states which may be present in awake animals. Furthermore, we report the stability and robustness of our imaging preparation in this study. We present the average of all the raw stimulation trials from each animal across the whole experimental session (*Figure 2—figure supplement 3*), showing the stability and robustness of our preparation, as well as easily identifying any changes. Future studies directly comparing stimulation-evoked haemodynamic responses in anaesthetised and awake PCSK9-ATH and J20-PCSK9-MIX mice would be useful in determining if such responses were still comparable under these diseased states. We have ensured that the level of anaesthesia is minimal and by examining vascular reactivity are able to ensure all animals are responsive to the same extent. Secondly, our imaging analysis assumes $O_2$ saturation to be 70% with a baseline haemoglobin concentration of 100 µM. This may be important if the assumed baselines are different in the diseased animals compared to WT controls; however, our recent study (*Sharp et al., 2020*) using the same J20-AD mouse model discussed this issue in detail, in which we showed that regardless of the baseline blood volume estimation used, our percentage change was scaled by it (i.e. always the same change). Therefore, the observations in this paper with respect to the different diseased animals are robust.

In conclusion, we report novel findings of impaired neurovascular function in a novel experimental model of atherosclerosis (PCSK9-ATH) characterised by reduced stimulus-evoked blood volume without any significant alterations to evoked neural activity showing evidence of neurovascular uncoupling. We induced atherosclerosis in a mild fAD model (J20-AD) to create a mixed comorbid model (J20-PCSK9-ATH) in which we report a significant increase in the number of hippocampal Aβ plaques, however, without any significant changes to evoked haemodynamic or neural responses compared to WT or J20-AD mice. A key finding from this study was CSD was more severe in diseased animals. This may reflect the global inflammatory state of the brain and could also serve to be an effective preclinical and human clinical biomarker for baseline state and to assess therapies. Future studies should include assessment of other inflammatory markers and cellular pathway changes by a genome wide transcriptomics approach from single cell populations. It would also be prudent to induce atherosclerosis in a more severe fAD model to provide a severity continuum of mixed models that reflect clinical presentations of dementia.

## Materials and methods
### Animals
All animal procedures were performed with approval from the UK Home Office in accordance to the guidelines and regulations of the Animal (Scientific Procedures) Act 1986 and were approved by the University of Sheffield ethical review and licensing committee. Male C57BL/6 J mice were injected i.v at 6 wks with $6 \times 10^{12}$ virus molecules/ml rAAV8-mPCSK9-D377Y (Vector Core, Chapel Hill, NC) and fed a Western diet (21% fat, 0.15% cholesterol, 0.03% cholate, 0.296% sodium; #829100, Special Diet Services UK) for 8 m (PCSK9-ATH). These mice were compared to age-matched wild-type C57BL/6 J mice (with no AAV injection fed normal rodent chow) that were used as controls (WT C57BL/6 J). In addition, male heterozygous transgenic (J20-hAPP B6.Cg-Zbtb20Tg(PDGFB-APPSwInd)20Lms/2Mmjax) (MMRRC Stock No: 34836-JAX) mice were used. Atherosclerosis was

induced in J20-hAPP mice alongside WT mice at 6 wks of age combined with a Western diet to create a comorbid mixed model (J20-PCSK9-MIX). For the CSD imaging experiments, 4 nNOS-ChR2 mice (M/F, 16–40 weeks old) were included in the WT group. [nNOS-ChR2 mice: heterozygous nNOS-CreER (Jax 014541, [*Taniguchi et al., 2011*]) x homozygous Ai32 mice (Jax 024109, [*Madisen et al., 2012*]), given tamoxifen (100 mg/kg, i.p., 3 injections over 5 days) at 1–2 months old]. All mice were imaged between 9–12 m of age. All mice were housed in a 12 hr dark/light cycle at a temperature of 23°C, with food and water supplied ad-libitum.

## Thinned cranial window surgery

Mice were anaesthetised with 7 ml/kg i.p. injection of fentanyl-fluanisone (Hypnorm, Vetapharm Ltd), midazolam (Hypnovel, Roche Ltd) and maintained in a surgical anaesthetic plane by inhalation of isoflurane (0.6–0.8% in 1 L/min $O_2$). Core body temperature was maintained at 37 °C through use of a homeothermic blanket (Harvard Apparatus) and rectal temperature monitoring. Mice were placed in a stereotaxic frame (Kopf Instruments, US) and the bone overlying the right somatosensory cortex was thinned forming a thinned cranial optical window. A thin layer of clear cyanoacrylate glue was applied over the cranial window to reinforce the window. Dental cement was applied around the window to which a metal head-plate was chronically attached. All mice were given 3 weeks to recover before the first imaging session.

## 2D-optical imaging spectroscopy (2D-OIS)

2D-OIS measures changes in cortical haemodynamics: total haemoglobin (HbT), oxyhaemoglobin (HbO) and deoxyhaemoglobin (HbR) concentrations (*Berwick et al., 2005*). Mice were lightly sedated and placed into a stereotaxic frame. Sedation was induced as described above and maintained using low levels of isoflurane (0.3–0.6%). For imaging, the right somatosensory cortex was illuminated using four different wavelengths of light appropriate to the absorption profiles of the differing haemoglobin states (494nm, 560nm, 575nm & 595nm) using a Lambda DG-4 high-speed galvanometer (Sutter Instrument Company, US). A Dalsa 1M60 CCD camera was used to capture the re-emitted light from the cortical surface. All spatial images recorded from the re-emitted light underwent spectral analysis based on the path length scaling algorithm (PLSA) as described previously (*Berwick et al., 2005*; *Mayhew et al., 1999*). which uses a modified Beer-Lambert law with a path light correction factor converting detected attenuation from the re-emitted light with a predicted absorption value. Relative HbT, HbR and HbO concentration estimates were generated from baseline values in which the concentration of haemoglobin in the tissue was assumed to be 100 μM and $O_2$ saturation to be 70%. For the stimulation experiments, whiskers were mechanically deflected for a 2s-duration and a 16s-duration at 5 Hz using a plastic T-shaped stimulator which caused a 1 cm deflection of the left-whisker. Each individual experiment consisted of 30 stimulation trials (for 2 s) and 15 stimulation trials (for 16 s) of which a mean trial was generated after spectral analysis of 2D-OIS. Stimulations were performed with the mouse breathing in 100% $O_2$ or 21% $O_2$, and a gas transition to medical air (21% $O_2$) as well as an additional 10% $CO_2$-hypercapnia test of vascular reactivity.

## Neural electrophysiology

Simultaneous measures of neural activity alongside 2D-OIS were performed in a final acute imaging session 1 week after the 1st imaging session. A small burr-hole was drilled through the skull overlying the active region (as defined by the biggest HbT changes from 2D-OIS imaging) and a 16-channel microelectrode (100 μm spacing, 1.5–2.7 MΩ impedance, site area 177 μm2) (NeuroNexus Technologies, USA) was inserted into the whisker barrel cortex to a depth of ~1500 μm. The microelectrode was connected to a TDT preamplifier and a TDT data acquisition device (Medusa BioAmp/RZ5, TDT, USA). Multi-unit analysis (MUA) was performed on the data. All channels were depth aligned to ensure we had 12 electrodes covering the depth of the cortex in each animal. The data were high pass filtered above 500 Hz to remove all low-frequency components and split into 100 ms temporal bins. Within each bin any data crossing a threshold of 1.5SD above the mean baseline was counted and the results presented in the form of fractional changes to MUA.

## Region analysis

Analysis was performed using MATLAB (MathWorks). An automated region of interest (ROI) was selected using the stimulation data from spatial maps generated using 2D-OIS. The threshold for a pixel to be included within the ROI was set at 1.5xSD, therefore the automated ROI for each session per animal represents the area of the cortex with the largest haemodynamic response, as determined by the HbT. For each experiment, the response across all pixels within the ROI was averaged and used to generate a time-series of the haemodynamic response against time.

## Statistical analysis

Statistical analyses were performed using SPSS v25, v26 and v27 and GraphPad Prism v8. Shapiro-Wilks test was used to check for normality and Levene's test was used to assess equality of variances. Two-way mixed design ANOVA, 1-way ANOVA or Kruskal-Wallis tests were used, as appropriate. For one-way ANOVA, if variances were unequal, Welch's F was reported. Results were considered statistically significant if $p < 0.05$. The Shapiro-Wilks test suggested that, for chronic experiments, peak values of HbT and HbO are normally distributed, however, HbR values are significantly non-normal. Two-way mixed design was used to compare peak values for HbT, HbO, and HbR (although HbR failed the S-W test for normality, an ANOVA was used as they were considered fairly robust against small deviations from normality). Inspection of Levene's test suggested that variances were equal, therefore, Dunnett's (two-sided) multiple comparisons test was used to compare disease models to WT, and for HbR, Games-Howell multiples comparisons were used. If the Greenhouse-Geisser estimate of sphericity showed deviation from sphericity (chronic experiments: HbT [$\varepsilon = 0.55$], HbO [$\varepsilon = 0.49$], and HbR [$\varepsilon = 0.564$]), results are reported with Greenhouse-Geisser correction applied. qRT-PCR data was analysed by performing one-way ANOVAs with Dunnett's multiple comparisons test used to compare disease models to WT. p-Values < 0.05 were considered statistically significant. All the data are presented as mean values ± standard error of mean (SEM).

## Immunohistochemistry

At the end of terminal experiments, mice were euthanised with an overdose of pentobarbital (100 mg/kg, Euthatal, Merial Animal Health Ltd) and transcardially perfused with 0.9% saline and brains were dissected. One half-hemisphere of the brains were fixed in formalin and embedded in paraffin wax, with the other half snap-frozen using isopentane and stored at –80°C. 5 µm coronal sections were obtained using a cryostat. Immunohistochemistry was performed using an avidin-biotin complex (ABC) method (as described previously [*Ameen-Ali et al., 2019*]). Following slide preparation and antigen retrieval (pressure cooker at 20 psi at 120 C for 45 s [pH6.5]), sections underwent additional pre-treatment in 70% formic acid. Sections were incubated with 1.5% normal serum followed by incubation with the primary antibody (biotinylated anti-Aβ – 1:100, BioLegend, USA) for 1 hr. Horseradish peroxidase avidin-biotin complex (Vectastain Elite Kit, Vector Laboratories, UK) was used to visualise antibody binding along with 3,3-diaminobenzidine tetrahydrochloride (DAB) (Vector Laboratories, UK). All sections were counterstained with haematoxylin, dehydrated and mounted in DPX. Sections were imaged using a Nikon Eclipse Ni-U microscope attached to a Nikon DS-Ri1 camera. Any well-defined brown dot of any size was considered a plaque. Plaques were identified at x40 magnification and manually counted per section.

## qRT-PCR

Snap-frozen hemispheres (brains dissected after terminal experiments as described above) were homogenised, and RNA was extracted using Direct-zol RNA MiniPrep kit with TRI-reagent as per the manufacturer's guidelines (Zymo) and RNA quality checked using NanoDropTM (ThermoFisher Scientific). cDNA was synthesised from the extracted RNA using the UltraScript 2.0 cDNA synthesis kit (PCR BioSystems) according to the manufacturer's guidelines. qRT-PCR was performed using PrimeTime qRT-PCR assay primers (IDT) for *IL1β* and *TNFα* with *ACTB* as the reference housekeeping gene. Luna qRT-PCR Master Mix (NEB) was used with the primers, cDNA and nuclease free water and each gene for each sample was duplicated. CFX384 Real-Time System (BioRad) with a C1000 Touch Thermal Cycler (BioRad) was used to perform qRT-PCR consisting of 40 cycles. Data was analysed using the well-established delta-Ct method (*Livak and Schmittgen, 2001*) by normalising against *ACTB*.

## Acknowledgements

OS's PhD studentship and consumables were funded by the Neuroimaging in Cardiovascular Disease (NICAD) network scholarship (University of Sheffield). A British Heart Foundation (BHF) project grant was awarded to SEF to carry out the work using the PCSK9 model (PG/13/55/30365). The J20-mouse colony was in part funded and supported by Alzheimer's Research UK (Grant R/153749-12-1). CH is funded by a Sir Henry Dale Fellowship jointly funded by the Wellcome Trust and the Royal Society. This research was funded in whole, or in part, by the Wellcome Trust [Grant number 105586/Z/14/Z]. For the purpose of Open Access, the author has applied a CC BY public copyright licence to any Author Accepted Manuscript version arising from this submission. MAR is funded by a Conacyt scholarship. We would like to thank Prof Lennart Mucke (Gladstone Institute of Neurological Disease & Department of Neurology, UCSF, CA, US) as well as the J David Gladstone Institutes for the J20-hAPP mice. Finally, we thank Mr Michael Port for building and maintaining the imaging apparatus and Dr Luke Boorman for producing MATLAB code for data analysis.

## Additional information

### Competing interests

The authors declare that no competing interests exist.

### Funding

| Funder | Grant reference number | Author |
| --- | --- | --- |
| University of Sheffield | Neuroimaging in Cardiovascular Disease (NICAD) network scholarship | Osman Shabir |
| British Heart Foundation | PG/13/55/30365 | Sheila E Francis |
| Wellcome Trust | 105586/Z/14/Z | Clare Howarth |
| Wellcome Trust | Sir Henry Dale Fellowship | Clare Howarth |
| Royal Society | Sir Henry Dale Fellowship | Clare Howarth |
| Alzheimer's Research UK | R/153749-12-1 | Jason Berwick |
| Consejo Nacional de Ciencia y Tecnología | | Monica A Rebollar |

The funders had no role in study design, data collection and interpretation, or the decision to submit the work for publication.

### Author contributions

Osman Shabir, Conceptualization, Data curation, Formal analysis, Investigation, Methodology, Writing – original draft; Ben Pendry, Llywelyn Lee, Formal analysis, Investigation; Beth Eyre, Paul S Sharp, Investigation, Writing – review and editing; Monica A Rebollar, Investigation; David Drew, Investigation, Resources; Clare Howarth, Data curation, Formal analysis, Funding acquisition, Investigation, Supervision, Writing – review and editing; Paul R Heath, Methodology, Supervision, Writing – review and editing; Stephen B Wharton, Conceptualization, Funding acquisition, Writing – review and editing; Sheila E Francis, Conceptualization, Funding acquisition, Methodology, Project administration, Supervision, Writing – review and editing; Jason Berwick, Conceptualization, Data curation, Formal analysis, Funding acquisition, Methodology, Project administration, Software, Supervision, Validation, Writing – review and editing

### Author ORCIDs

Osman Shabir http://orcid.org/0000-0001-7412-6966
Llywelyn Lee http://orcid.org/0000-0003-3449-9797
Clare Howarth http://orcid.org/0000-0002-6660-9770

## Ethics

All animal procedures were performed with approval from the UK Home Office in accordance to the guidelines and regulations of the Animal (Scientific Procedures) Act 1986 and were approved by the University of Sheffield ethical review and licensing committee.

## Decision letter and Author response

Decision letter https://doi.org/10.7554/eLife.68242.sa1
Author response https://doi.org/10.7554/eLife.68242.sa2

## Additional files

### Supplementary files

• Transparent reporting form

### Data availability

Data files including figure generating MATLAB code and numerical values used for statistical analyses (Excel and SPSS) have been uploaded to Dyrad (doi: https://doi.org/10.5061/dryad.z08kprrcj).

The following dataset was generated:

| Author(s) | Year | Dataset title | Dataset URL | Database and Identifier |
|---|---|---|---|---|
| Waggitt J | 2021 | Data from: Assessment of Neurovascular Coupling & Cortical Spreading Depression in Mixed Mouse Models of Atherosclerosis and Alzheimer's Disease | http://dx.doi.org/10.5061/dryad.z08kprrcj | Dryad Digital Repository, 10.5061/dryad.z08kprrcj |

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
