## [Editor Report]

In their manuscript, Shabir et al., examine changes in cerebral neurovascular coupling in mouse models of familial Alzheimer disease, atherosclerosis, and a combined comorbidity model to determine the impact of Alzheimer's disease and arteriosclerosis comorbidity on neurovascular coupling. The authors report a set of observations derived from intrinsic optical imaging and multi unit recordings performed in these mouse lines under different combinations of stimulus length and partial oxygen pressure. The discovery that both sensory-driven and injury-based changes in cerebral blood flow (CBF) are perturbed in the settings of Alzheimer's disease and atherosclerosis will help to understand how these diseases impair neurovascular coupling.

---

## [Decision Letter]

**Decision letter after peer review:**

Thank you for submitting your article "Assessment of Neurovascular Coupling and Cortical Spreading Depression in Mixed Models of Atherosclerosis and Alzheimer's Disease" for consideration by *eLife*. Your article has been reviewed by 2 peer reviewers, and the evaluation has been overseen by a Reviewing Editor and a Senior Editor. The reviewers have opted to remain anonymous.

The reviewers have discussed their reviews with one another, and the Reviewing Editor has drafted this letter to help you prepare a revised submission.

Essential revisions:

The Reviewers pointed out that all of the points raised in their comments are considered essential revisions and to be addressed with additional data and analyses.*Reviewer #1:*

Strengths:

The data presented in the paper is of pristine quality and the experimental setting has the power to uncover the changes in neurovascular coupling in a novel model of Alzheimer's disease and arteriosclerosis comorbidity; making this work particularly attractive and interesting.

Weaknesses:

1. Neurovascular "coupling" is not directly assessed and more detailed analysis of the hemodynamics is required to contrast between groups.

2. The paper is largely descriptive and no mechanistic insight is provided.

3. Some interpretation of the data ignores trends in the data (even if not statistically significant), that could lead to different conclusion if group sizes were larger or data analyzed differently.

4. The authors claim that the exaggerated HbT response to CSD in diseased animals results in increased hypoxia and ischemia yet no direct evidence is provided to support this claim.

5. Potential differential interactions between anesthetic agent with neuronal and vascular responses between experimental groups is not addressed.

Discussed here below are the points raised by this reviewer that need to be addresses to strengthen the claims made in the manuscript; they are elaborated in the same order as presented in the "Weakness" section above.

1. The most fundamental message that the authors aim to deliver in the paper is the abnormal neurovascular coupling, particularly in the MIX model. The conclusion is largely based on the fact that the WT mice had a significantly larger peak HbT amplitude (Figure 2A and B), this is just one metric that can be extracted from these time series. For example, there seems to be no differences in onset time or time to peak (obviously a rigorous analysis is required). On the flip side, a close-up look at the initial phase of the HbR shows that, for the WT, there is an almost immediate decrease in the fractional change while the other models show a more delayed (even opposite) directional change, particularly for the 2s stimulation under 100% and 21% oxygen. It should be notice that the shorter the stim, the more realistic it becomes given the nature of the whisker system. Even if one focuses solely on the HbT amplitude changes, how do these correlate with the underlying neuronal activity? Looking at the data presented in Figure 4, one can observe a clear trend (not statistically significant by the current analysis) where the activity in the WT has also a larger amplitude. A scatter plot of the peak MUA change vs peak fractional HbT for each trial could prove informative while exploring this questions. Next one would like to test and see if the regression coefficient between these variable changes between the different models and if so, what is the measured difference (i.e. changes in regression slope between groups). Not all spectral bands are equally correlated with haemodynamics so one would need to repeat this analysis for representative bands. Bottom line, there is a good chance to find some hidden gems in this high-quality data and it worths to dive into a more detailed analysis.

2. Alas, by "mechanistic" this reviewer does not implies finding a specific molecular pathway. On the contrary, in the context of the paper, a mechanistic explanation of the results could have been focused on differential responses across specific vascular compartments. For example, uncovering the vascular compartment behind the difference in HbT; one would hypothesize that pial arteries and the penetrating arteries covered by smooth-muscle cells might be the culprit as those are expected to loose functional responses as they become stiffer in arteriosclerosis ,similar to hypertension. Measuring changes in vessel diameter across pial and penetrating vessels (arteries and veins) would have pointed into a potential explanation for the observed phenomena. The authors should also measure changes in capillary structure (i.e. diameter, partial volume etc). Further, a detailed histological survey of the vascular compartments (arterial lumen vs wall diameter), pericyte/smooth-muscle coverage, could have been much more relevant than neuropathology and neuroinflammation data (Figure 5) which do not contribute directly to the neurovascular narrative of the manuscript. Moreover, the vascular and neuronal data are recorded from the cortex where the authors find no pathology hence the hippocampal data is harder to connect to functional findings presented in the manuscript. That said, the increased plaque density in the MIX model is intriguing raising questions regarding aBeta clearing mechanism; testing if this model recapitulates impaired clearing such in hypertension if of much interest albeit might be set aside for another manuscript.

3. Although rigorous statistics should be our guidance wile navigating the obscure waters of hypothesis testing, one cannot avoid but observing trends present in the data. Such observation might have been even used in a grant application as preliminary data, in the sake of asking for additional fundings to hone into the study of an intriguing phenomenon. I found that there are several of these cases that call for mentioning, as the conclusion of this manuscript might have differed if found to be statistically significant. First, I would like to point to the hypercapnia challenge presented in S3. In the chronic conditions (left) the response in the WT appears larger than in the other groups. While there where no statistical differences in peak values, it will be more representative to look at Area Under the Curve (AUC) as it might provide a more robust metric for comparison. A similar trend (mentioned above) can be seen for the multi-unit recording (Figure 4), WT has a larger amplitude in three out of four conditions. It would have been important to discuss, at least, what was the minimal effect size that could have been discovered given the group sizes and measured variance, or how many more animals would have been needed to detect a statistically significant difference for the observed means. It would have been instructive to have additional panels with summary statistics (such as peak amplitude) showing trial-by-trial variability.

4. As pointed by the authors, it is indeed expected that reduced HbT might lead to more pronounced ischemia and hypoxia and the tools to demonstrate are available calling to, at least, a preliminary demonstration that this is indeed the case. It should be noted that one should also expect to see already an increased basal level, at least for hypoxia markers, in the AD and MIX models.

5. The authors point in the discussion to the unavoidable limitations imposed by performing this line of research in anesthetized animals, their relay on previous work from their laboratory showing the specific anesthesia protocol is comparable to awake imaging in terms of vascular responses to whisker stimulation. It is imperative to show that this assumed lack of difference between awake and anesthetized conditions holds true for the AD and MIX models. This reviewer is aware of the complexity of such demonstration as vascular reactivity is expected to be impaired in arteriosclerosis conditions. Similarly, it is fundamental to rule out a potential interaction between the different mouse models and the response to the anesthetic agent in terms of neuronal activity. There is certainly much to debate regarding the optimal experimental setup (light anesthesia vs awake), pros and cons exist for both. We (as experimenters) should aim to make a clear statement about the impact of the experimental conditions on the metrics we report to better interpret our results. While reproducing the same experiment under awake conditions is beyond the scope of this work, we need to make sure vascular physiology and neuronal responses are not differentially affected in each experimental group.

*Reviewer #2:*

Summary:

Impairments in vascular structure/function likely play a causal role in the cognitive decline associated with aging and neurodegenerative disease. The purpose of this study (by the Berwick group) was to examine changes in cerebral neurovascular coupling in animal models of familial Alzheimer disease (J20 mouse), atherosclerosis (mPCSK9) and a combined comorbidity model. The authors make the novel discovery that both sensory driven and injured based changes in cerebral blood flow (CBF) are perturbed in the atherosclerosis model.

Strengths:

The authors are recognized experts in the methodologies employed, and not surprisingly, the experiments were conducted with rigor. The study incorporates multiple middle aged animal models to study disease related changes in vascular function. Of particular significance is the inclusion of age/disease related co-morbidities such as the atherosclerosis model. As a result, they made the novel discovery that atherosclerosis impairs the regulation of CBF. Overall, I think this is a well executed study whose primary conclusions were mostly justified by the data presented. Although I think this study adds important new knowledge, some additional analyses and changes to data/figure presentation would really enhance the readability of the manuscript.

Weaknesses:

One potential weakness could be that since mice were assessed under anesthesia in both 100% and 21% oxygen (hyperoxia and normoxia) , one wonders if there were baseline differences in cardiovascular parameters (blood oxygenation, BP, HR etc) that could partially explain why cerebral blood flow responses were altered. If baseline blood oxygenation levels were different between animal models (perhaps elevated under normoxic conditions), there may be less need to recruit blood flow to active cortical areas. The authors describe this limitation in the discussion but there are ways to address this issue in atherosclerotic mice, pulse oximetry for example.

1) Regarding the analysis. I found the paper makes the reader work really hard to understand the comparisons, analyses and effects. The data presented in figure 2, 3 and 4 show complex waveforms for each component of HbO, HbT and HbR in each animal model and each experimental condition (eg. Figure 2: 16 specific conditions x 3 waveforms per condition = 48 waveforms presented). However, it was not completely clear to me how these waveforms were analysed and compared across groups. It appears that just peak value was assessed but it left me wondering if other metrics such as total area under the curve, latency, or slope should also considered (maybe they don't have value?). In some cases the polarity of the waveform is bi or tri-phasic, is this aspect accounted for in the analyses? For the readers' sake, could the authors create graphs in Figure 2-4 that effectively captures how these waveforms were compared across groups and statistically analysed (eg. bar graph with individual data points or whisker box plot of peak amplitudes for each group)?

2) One result which confused me was the statement that ATH mice show a significant reduction in HbT, whereas the figure legend 2 starts with the statement that there was no significant overall effect of disease. Since it is clear that the different disease groups were going to be compared against WT mice anyways (a priori planned comparisons), I didn't understand why one would bother with the non-significant omnibus test to begin with.

3) Figure 3, how fast was the electrode inserted relative to the times shown in Figure 3A? Was the electrode still being pushed into the brain 20-60 seconds after touching the pial surface? Is this what the sharp deflection in values represent in Figure 3B? One small request would be to please note this aspect in the Figure. In addition, the main group differences appear to be from ~150-2000 seconds, perhaps the images shown in 3A could be extended to include more of the data beyond 150s in the experiment.

4) Does the mechanical deformation of the brain caused by the electrode insertion (I would expect the dura would first indent, then give way as the electrode pierces it) artificially change the reflectance of different wavelengths off the surface? A control could be to image during electrode insertion in a dead animal or one with brain activity silenced. Secondly, how do the authors know that the electrode induced a spreading depression wave, since that is typically shown with DC recordings? Perhaps this is already known from previous studies, but it should be mentioned in the results with an appropriate citation.

5) I think the authors should spend more text space describing the finding that sensory evoked responses were not different across any disease model 35 min after the electrode was inserted (Supp Figure 1). To help the reader, what does this finding mean in the broader context? Does it conflict at all with the findings presented in Figure 2? Does it reflect the fact that several metrics had not returned to baseline even 35min after electrode insertion?

6) Figure 4, the authors state that neural activity wasn't significantly different between groups but based on area under the curve for fractional changes in activity, there appears to be noticeably more activity in WT mice across the 2 or 16 sec stimulation duration (first 2 panels). Is this aspect not considered in the analysis? If not, it would be helpful to provide a rationale for why.

7) For mRNA analysis of inflammation, it was not clear under what conditions tissue was extracted. Was tissue extracted after an experiment (sticking an electrode in the brain) or in naïve animals?

---

## [Author Response]

Essential revisions:The Reviewers pointed out that all of the points raised in their comments are considered essential revisions and to be addressed with additional data and analyses.Reviewer #1:Strengths:The data presented in the paper is of pristine quality and the experimental setting has the power to uncover the changes in neurovascular coupling in a novel model of Alzheimer's disease and arteriosclerosis comorbidity; making this work particularly attractive and interesting.

We would like to thank the reviewer for their kind comments and recognition that this is an important issue, and the experimental work is a high standard.

Weaknesses:1. Neurovascular "coupling" is not directly assessed and more detailed analysis of the hemodynamics is required to contrast between groups.2. The paper is largely descriptive and no mechanistic insight is provided.3. Some interpretation of the data ignores trends in the data (even if not statistically significant), that could lead to different conclusion if group sizes were larger or data analyzed differently.4. The authors claim that the exaggerated HbT response to CSD in diseased animals results in increased hypoxia and ischemia yet no direct evidence is provided to support this claim.5. Potential differential interactions between anesthetic agent with neuronal and vascular responses between experimental groups is not addressed.

We would like to thank the reviewer for highlighting some of the weaknesses and in response to the weaknesses highlighted, we have addressed all the issues (points 1-4) which are detailed below.

Discussed here below are the points raised by this reviewer that need to be addresses to strengthen the claims made in the manuscript; they are elaborated in the same order as presented in the "Weakness" section above.1. The most fundamental message that the authors aim to deliver in the paper is the abnormal neurovascular coupling, particularly in the MIX model. The conclusion is largely based on the fact that the WT mice had a significantly larger peak HbT amplitude (Figure 2A and B), this is just one metric that can be extracted from these time series. For example, there seems to be no differences in onset time or time to peak (obviously a rigorous analysis is required). On the flip side, a close-up look at the initial phase of the HbR shows that, for the WT, there is an almost immediate decrease in the fractional change while the other models show a more delayed (even opposite) directional change, particularly for the 2s stimulation under 100% and 21% oxygen. It should be notice that the shorter the stim, the more realistic it becomes given the nature of the whisker system. Even if one focuses solely on the HbT amplitude changes, how do these correlate with the underlying neuronal activity? Looking at the data presented in Figure 4, one can observe a clear trend (not statistically significant by the current analysis) where the activity in the WT has also a larger amplitude. A scatter plot of the peak MUA change vs peak fractional HbT for each trial could prove informative while exploring this questions. Next one would like to test and see if the regression coefficient between these variable changes between the different models and if so, what is the measured difference (i.e. changes in regression slope between groups). Not all spectral bands are equally correlated with haemodynamics so one would need to repeat this analysis for representative bands. Bottom line, there is a good chance to find some hidden gems in this high-quality data and it worths to dive into a more detailed analysis.

We thank the reviewer for their kind comments and recognition that this is an important area of research. We agree

We have undertaken more data analysis as suggested. We have analysed the data for the arterial region as well as for the “area under curve” and included that in the supplemental data.

We have included rise times and times to peak data in the supplemental data, confirming there are no differences in these metrics across any of the groups or conditions.

In response to both reviewers’ comments concerning comparison of HbR between groups, we have added the following sentence in the manuscript:

*“*Examining the shape of the HbR washout in the some of the disease models shows key differences compared to WT mice. Most notably, J20-PCSK9-MIX mice show a triphasic response with an initial increase in HbR followed by a decrease then return to baseline (Figure 2). This contrasts with the biphasic waveform seen in WT, J20-AD and PCSK9-ATH mice; albeit lower in disease mice compared to WT controls.”

We agree that neurovascular coupling was not directly assessed despite having measures for both evoked haemodynamics and neural activity. We have therefore produced a neurovascular correlation scatter plot (shown in Figure 4; 4B) comparing the evoked MUA vs evoked HbT in the 4 different mouse groups in both the first 2 stimulations under oxygen and the second 2 stimulations under air. We find that the regression coefficient is poorer for atherosclerotic mice compared to WT as for the same amount of MUA there is less HbT suggesting neurovascular uncoupling. This finding confirms our fundamental message that atherosclerotic mice have significantly impaired haemodynamics compared to WT mice.

2. Alas, by "mechanistic" this reviewer does not implies finding a specific molecular pathway. On the contrary, in the context of the paper, a mechanistic explanation of the results could have been focused on differential responses across specific vascular compartments. For example, uncovering the vascular compartment behind the difference in HbT; one would hypothesize that pial arteries and the penetrating arteries covered by smooth-muscle cells might be the culprit as those are expected to loose functional responses as they become stiffer in arteriosclerosis ,similar to hypertension. Measuring changes in vessel diameter across pial and penetrating vessels (arteries and veins) would have pointed into a potential explanation for the observed phenomena. The authors should also measure changes in capillary structure (i.e. diameter, partial volume etc). Further, a detailed histological survey of the vascular compartments (arterial lumen vs wall diameter), pericyte/smooth-muscle coverage, could have been much more relevant than neuropathology and neuroinflammation data (Figure 5) which do not contribute directly to the neurovascular narrative of the manuscript. Moreover, the vascular and neuronal data are recorded from the cortex where the authors find no pathology hence the hippocampal data is harder to connect to functional findings presented in the manuscript. That said, the increased plaque density in the MIX model is intriguing raising questions regarding aBeta clearing mechanism; testing if this model recapitulates impaired clearing such in hypertension if of much interest albeit might be set aside for another manuscript.

We appreciate that we do not provide a specific molecular mechanism aside from neuroinflammation and work on this aspect will be the focus of future papers. The interesting mechanistic aspects mentioned by the reviewer are a programme of work all by themselves. At this stage, we wished to present the models and the exciting and detailed neurophysiological findings associated with them to the research community. The focus of our study was to characterise the ATH and MIX models, as compared to both WT and AD, with detailed analysis of the underlying mechanisms, molecular and cellular changes, being beyond the scope of the current study. Such studies (e.g. collagen IV staining of vessel thickness and density will form the basis of future work). We agree with the reviewer that looking at capillary structure in vivo is an interesting area of research, but it is not possible with our optical imaging setup used in this study but should from the basis for future studies also. We have, however, now included data from specific arterial ROIs (as pointed out in response to the above point) in the supplemental information. With respect to the neuroinflammation data, this is an area we have published on previously (Denes et al., JAHA 2012 J Am Heart Assoc 2012 Jun;1(3):e002006. doi: 10.1161/JAHA.112.002006). and we wanted to extend our findings to these new models. Much more detailed analysis of molecular and cellular changes will be the basis of future manuscripts.

With respect to amyloid β staining, we provided the significant differences (hippocampal) as there was not a clear increase within the cortex at this timepoint. However, if you combine whole brain amyloid β then there is a significant increase overall and this has now been reported in a Table 1 in the supplemental information.

3. Although rigorous statistics should be our guidance wile navigating the obscure waters of hypothesis testing, one cannot avoid but observing trends present in the data. Such observation might have been even used in a grant application as preliminary data, in the sake of asking for additional fundings to hone into the study of an intriguing phenomenon. I found that there are several of these cases that call for mentioning, as the conclusion of this manuscript might have differed if found to be statistically significant. First, I would like to point to the hypercapnia challenge presented in S3. In the chronic conditions (left) the response in the WT appears larger than in the other groups. While there where no statistical differences in peak values, it will be more representative to look at Area Under the Curve (AUC) as it might provide a more robust metric for comparison. A similar trend (mentioned above) can be seen for the multi-unit recording (Figure 4), WT has a larger amplitude in three out of four conditions. It would have been important to discuss, at least, what was the minimal effect size that could have been discovered given the group sizes and measured variance, or how many more animals would have been needed to detect a statistically significant difference for the observed means. It would have been instructive to have additional panels with summary statistics (such as peak amplitude) showing trial-by-trial variability.

We thank the reviewer for their suggestion and have investigated the area under curve data for chronic hypercapnia (see supplemental information). Using this metric, we found no significant differences between groups, confirming our conclusion that there are no differences in vascular reactivity between disease models. These findings support our suggestion that all differences are due to functional, rather than vascular deficits.

We have added summary statistics in a table in the supplemental information (Table 1).

4. As pointed by the authors, it is indeed expected that reduced HbT might lead to more pronounced ischemia and hypoxia and the tools to demonstrate are available calling to, at least, a preliminary demonstration that this is indeed the case. It should be noted that one should also expect to see already an increased basal level, at least for hypoxia markers, in the AD and MIX models.

We do not currently have measures of hypoxia directly, but these will be explored on a cellular and molecular level in future publications. We have added the following sentence to the manuscript:

“We would also expect to see increased basal levels of ischaemia and hypoxia to some extent in the disease models, particularly the mixed model. Investigation of such markers will form the basis of future studies.”

5. The authors point in the discussion to the unavoidable limitations imposed by performing this line of research in anesthetized animals, their relay on previous work from their laboratory showing the specific anesthesia protocol is comparable to awake imaging in terms of vascular responses to whisker stimulation. It is imperative to show that this assumed lack of difference between awake and anesthetized conditions holds true for the AD and MIX models. This reviewer is aware of the complexity of such demonstration as vascular reactivity is expected to be impaired in arteriosclerosis conditions. Similarly, it is fundamental to rule out a potential interaction between the different mouse models and the response to the anesthetic agent in terms of neuronal activity. There is certainly much to debate regarding the optimal experimental setup (light anesthesia vs awake), pros and cons exist for both. We (as experimenters) should aim to make a clear statement about the impact of the experimental conditions on the metrics we report to better interpret our results. While reproducing the same experiment under awake conditions is beyond the scope of this work, we need to make sure vascular physiology and neuronal responses are not differentially affected in each experimental group.

We completely agree with the reviewer about their comments regarding anaesthesia and we take the aspects mentioned very seriously indeed. Although these suggestions are not in the scope of the present study we have added the following sentences in the manuscript:

“Additional studies comparing anaesthetised and awake responses would be useful in determining if such responses were still comparable under diseased states. We have ensured that the level of anaesthesia is minimal and by examining vascular reactivity are able to ensure all animals are responsive to the same extent”.

Reviewer #2:Summary:Impairments in vascular structure/function likely play a causal role in the cognitive decline associated with aging and neurodegenerative disease. The purpose of this study (by the Berwick group) was to examine changes in cerebral neurovascular coupling in animal models of familial Alzheimer disease (J20 mouse), atherosclerosis (mPCSK9) and a combined comorbidity model. The authors make the novel discovery that both sensory driven and injured based changes in cerebral blood flow (CBF) are perturbed in the atherosclerosis model.Strengths:The authors are recognized experts in the methodologies employed, and not surprisingly, the experiments were conducted with rigor. The study incorporates multiple middle aged animal models to study disease related changes in vascular function. Of particular significance is the inclusion of age/disease related co-morbidities such as the atherosclerosis model. As a result, they made the novel discovery that atherosclerosis impairs the regulation of CBF. Overall, I think this is a well executed study whose primary conclusions were mostly justified by the data presented. Although I think this study adds important new knowledge, some additional analyses and changes to data/figure presentation would really enhance the readability of the manuscript.

We thank the reviewer for the very kind comments especially about the rigor of the conducted experiments as we take this very seriously.

Weaknesses:One potential weakness could be that since mice were assessed under anesthesia in both 100% and 21% oxygen (hyperoxia and normoxia) , one wonders if there were baseline differences in cardiovascular parameters (blood oxygenation, BP, HR etc) that could partially explain why cerebral blood flow responses were altered. If baseline blood oxygenation levels were different between animal models (perhaps elevated under normoxic conditions), there may be less need to recruit blood flow to active cortical areas. The authors describe this limitation in the discussion but there are ways to address this issue in atherosclerotic mice, pulse oximetry for example.

We would like to thank the reviewer for highlighting some potential issues in our analysis and discussion. We have addressed these comments in detail below and we hope to increase the clarity of the analysis undertaken.

1) Regarding the analysis. I found the paper makes the reader work really hard to understand the comparisons, analyses and effects. The data presented in figure 2, 3 and 4 show complex waveforms for each component of HbO, HbT and HbR in each animal model and each experimental condition (eg. Figure 2: 16 specific conditions x 3 waveforms per condition = 48 waveforms presented). However, it was not completely clear to me how these waveforms were analysed and compared across groups. It appears that just peak value was assessed but it left me wondering if other metrics such as total area under the curve, latency, or slope should also considered (maybe they don't have value?). In some cases the polarity of the waveform is bi or tri-phasic, is this aspect accounted for in the analyses? For the readers' sake, could the authors create graphs in Figure 2-4 that effectively captures how these waveforms were compared across groups and statistically analysed (eg. bar graph with individual data points or whisker box plot of peak amplitudes for each group)?

We have produced bar graphs of the peak HbT, HbO and HbR for each of the conditions alongside Figure 2. Similarly, we have added bar graphs showing MUA peak in Figure 4.

We have investigated area under curve values for both haemodynamic and neural MUA data and presented these in the supplemental data, as mentioned in our response to Reviewer 1, above.

With regards to response latency, we have investigated time to peak and rise times and have included graphs for these in the supplemental information. We found no differences in these metrics across any of the groups or conditons.

We have added the following sentence describing the waveforms for HbR in the manuscript: “Examining the shape of the HbR washout in the some of the disease models shows key differences compared to WT mice. Most notably, J20-PCSK9-MIX mice show a triphasic response with an initial increase in HbR followed by a decrease then return to baseline (Figure 2). This contrasts with the biphasic waveform seen in WT, J20-AD and PCSK9-ATH mice; albeit lower in disease mice compared to WT controls.”

2) One result which confused me was the statement that ATH mice show a significant reduction in HbT, whereas the figure legend 2 starts with the statement that there was no significant overall effect of disease. Since it is clear that the different disease groups were going to be compared against WT mice anyways (a priori planned comparisons), I didn't understand why one would bother with the non-significant omnibus test to begin with.

Our main comparison of interest was to compare disease groups to WT, as pointed out by the reviewer. However, for HbO and HbR, the omnibus test revealed a main effect of disease, therefore, we have included the omnibus test results for completeness. For HbT, Dunnett’s test results are sufficient to show the differences in ATH mice compared to WT as the research question was are any of the disease groups different to WT.

3) Figure 3, how fast was the electrode inserted relative to the times shown in Figure 3A? Was the electrode still being pushed into the brain 20-60 seconds after touching the pial surface? Is this what the sharp deflection in values represent in Figure 3B? One small request would be to please note this aspect in the Figure. In addition, the main group differences appear to be from ~150-2000 seconds, perhaps the images shown in 3A could be extended to include more of the data beyond 150s in the experiment.

Insertion of the electrode into the brain occurs at the 0s timepoint, depicted in Figure 3A and is completed within 5s. Therefore, the electrode was not being pushed into the brain 20-60s after touching the pial surface and so the sharp deflection in Figure 3B reflects genuine decreases in HbT caused by CSD (characteristic profile, discussed in more detail in point 4 below). We thank the reviewer for raising this point and have added further information in the figure legend of Figure 3A to clarify this point.

We have now extended the image montage to include spatial figures beyond 150s and also extended this figure to include representative images from ATH and MIX mice.

4) Does the mechanical deformation of the brain caused by the electrode insertion (I would expect the dura would first indent, then give way as the electrode pierces it) artificially change the reflectance of different wavelengths off the surface? A control could be to image during electrode insertion in a dead animal or one with brain activity silenced. Secondly, how do the authors know that the electrode induced a spreading depression wave, since that is typically shown with DC recordings? Perhaps this is already known from previous studies, but it should be mentioned in the results with an appropriate citation.

We do not make a region of interest where the electrode site is. Our ROIs do not include the electrode site itself as such all data is not artificially changed by the electrode being present or any artificial changes in reflectance due to deformation of brain tissue.

The haemodynamic profile which we observe on electrode insertion is characteristic of CSD. The characteristic haemodynamic profiles that occur during CSD are well known and the following citations describe these further. We have added these citations to the text.

Østergaard et al., 2015. Neurovascular coupling during cortical spreading depolarization and -depression. Stroke. 46(5):1392-401.

Sanchez-Porras et al., 2017. Ketamine modulation of the haemodynamic response to spreading depolarization in the gyrencephalic swine brain. J Cereb Blood Flow Metab. 37(5):1720–1734.

Sun et al., 2011. Simultaneous monitoring of intracellular pH changes and hemodynamic response during cortical spreading depression by fluorescence-corrected multimodal optical imaging. Neuroimage. 57(3):873-84.

Chang et al., 2010. Biphasic direct current shift, haemoglobin desaturation and neurovascular uncoupling in cortical spreading depression. Brain. 133(Pt 4):996-1012.

5) I think the authors should spend more text space describing the finding that sensory evoked responses were not different across any disease model 35 min after the electrode was inserted (Supp Figure 1). To help the reader, what does this finding mean in the broader context? Does it conflict at all with the findings presented in Figure 2? Does it reflect the fact that several metrics had not returned to baseline even 35min after electrode insertion?

Confounds of CSD mean that haemodynamics are altered and thus may be different in acute imaging sessions as compared to chronic imaging sessions. As such, haemodynamic responses in chronic and acute experiments are different initially where chronic better reflects the “true” reflection of disease-related haemodynamics, whereas acute sessions reflect the addition of mild trauma caused by electrode insertion in combination with disease where the CSD is bigger. We have added the following sentences to the manuscript:

“This is due to the effect CSD has on baseline and evoked haemodynamics that persist for some time up to and during the experimental stimulations. Thus, the acute data presents a different outlook compared to chronic-only imaging sessions that do not have the added confound of CSD.”

6) Figure 4, the authors state that neural activity wasn't significantly different between groups but based on area under the curve for fractional changes in activity, there appears to be noticeably more activity in WT mice across the 2 or 16 sec stimulation duration (first 2 panels). Is this aspect not considered in the analysis? If not, it would be helpful to provide a rationale for why.

We thank the reviewer for their suggestion and we have now compared area under the curve for fractional changes in multi-unit activity. We do not find any significant differences in MUA using this metric across any of the groups across any of the stimulations. We have reported this finding in Table 1 of the supplemental information.

7) For mRNA analysis of inflammation, it was not clear under what conditions tissue was extracted. Was tissue extracted after an experiment (sticking an electrode in the brain) or in naïve animals?

We have now clarified in the methods that for mRNA analysis, brains were dissected after terminal experiments. Thus, all brains are from the experimental mice rather than naïve ones, allowing neurophysiological and neuropathological findings to be related directly within the same mice.